# Nitrogen and Phosphorus Release Characteristics of Pipeline Sediments on Entering Different Water Bodies

**Jiarong Sun [1], Chonghua Xue [1,*], Junqi Li [1,2] and Wenhai Wang [1]**

[1] Key Laboratory of Urban Stormwater System and Water Environment, Ministry of Education, Beijing University of Civil Engineering and Architecture, Beijing 100044, China; 2108570020053@stu.bucea.edu.cn (J.S.); lijunqi@bucea.edu.cn (J.L.); wangwenhai@bucea.edu.cn (W.W.)

[2] Beijing Energy Conservation & Sustainable Urban and Rural Development Provincial and Ministry Co-Construction Collaboration Innovation Center, Beijing 100044, China

[*] Correspondence: xuechonghua@bucea.edu.cn

**Abstract:** Differences in the physical and chemical properties of reclaimed water (RW) and natural surface water (SW) lead to further differences in nitrogen and phosphorus release when pipeline sediments enter these water bodies. The release kinetics of nitrogen and phosphorus from pipe sediments with different particle sizes have been investigated. The results demonstrated that both SW and RW had a pH buffering effect after sediment addition, and the final pH (approximately 8.1) of RW was lower. The release of total phosphorus (TP) and ammonia nitrogen ($NH_4^+$-N) fitted the first-order kinetic model in which the release of TP reached equilibrium. TP release was inhibited in both SW and RW, with RW exhibiting the lowest (by a factor of 1.23~2.44) release (0.002 mg/g). The release of $NH_4^+$-N was promoted in both SW and RW; the maximum release in RW was 0.0188 mg/g. The amounts of $NH_4^+$-N released in SW and RW were 1.02–1.40 and 1.30–1.80 times that of the control group (CG), respectively. The percentage of TP and $NH_4^+$-N release in the three groups was highest in 75–154 μm pipe sediment, reaching 34.53% and 43.51% in SW and RW, respectively. These results can assist in the development of water quality evolution models for specific urban scenarios and provide important guidance for the precise regulation of water recharge quality during and after rainfall.

**Keywords:** pipeline sediment; reclaimed water; surface water; release of TP and $NH_4^+$-N

## 1. Introduction

In recent years, eutrophication has become a rapidly intensifying global environmental crisis [1]. Organic pollutants, including nitrogen and phosphorus, can enter urban water bodies through stormwater pipe runoff and combined sewer overflow and are a major cause of eutrophication [2,3]. Pipeline sediment serves as the principal carrier in the migration and transformation of nitrogen and phosphorus pollutants. The accumulation of nitrogen and phosphorus pollutants in pipes is intensified by rainwater runoff deposits containing particulate matter, and the sediments in rainwater pipes are a sink for nitrogen and phosphorus pollution [4]. During heavy rainfall, the sediments in storm water pipes are discharged into the downstream water body where erosion combined with pipeline runoff releases significant quantities of nitrogen and phosphorus [5,6]. A previous study has established that the contribution of pipeline sediment to the nitrogen and phosphorus pollution load in runoff is approximately 30~40% during rainfall events [7]. Moreover, the contribution of pipeline sediments to the organic pollution load is as high as 80% in rainstorm overflow events [8]. Therefore, it is crucial to explore the release of nitrogen and phosphorus when pipeline sediments enter the water body.

With the acceleration of urbanization and the requirement for the implementation of sustainable strategies, urban water environment governance has assumed increasing importance. Due to the lack of available surface water resources in some cities, urban river channels often dry up. In order to maintain the landscape of urban rivers and lakes

and restore the ecological balance, many cities are now actively considering alternative water resources, and reclaimed water is a significant low-cost source of ecological water that can replenish urban rivers and lakes [9]. In Beijing, reclaimed water has become the second most stable water source, and the proportion of reclaimed water that replenishes rivers and lakes has increased from 51.92% in 2012 to 90.97% in 2019 [10]. However, urban landscape water that is supplied by reclaimed water is prone to eutrophication problems, such as blooms and red tides, due to shallow depths, slow flow rates, and high concentrations of nitrogen and phosphorus contaminants [11,12]. Eutrophication problems typically increase turbidity and reduce water clarity and dissolved oxygen (DO), which ultimately results in cloudy water with an unpleasant odor. The deterioration of water quality not only reduces the landscape effect but also poses a risk to water ecosystems and human health [13]. Meanwhile, environmental factors, such as pH and dissolved oxygen, in reclaimed water recharge are also significantly different from those in natural water [14]. Relevant studies have shown that these differences impact the adsorption, accumulation and release of nitrogen and phosphorus by particulate matter in rainfall runoff [15]. Therefore, in urban watercourses with reclaimed water from sewage plants as the main recharge source, the migration and transformation patterns of rainfall runoff pollutants differ from those in natural watercourses. The application of previous research results and water quality models to urban watercourses with a specific recharge source can result in inaccurate predicted trends with respect to the water environment during and after rainfall. Therefore, it is of immediate practical value to study the transformation mechanism of rainfall runoff pollutants in urban river channels, establishing water quality evolution models that conform to specific urban circumstances and allow the effective regulation of river water quality during and after rainfall to maintain high urban water environment quality.

Studies on the release of nitrogen and phosphorus from sediments have focused on road sediments and the sediment–water interfaces of natural rivers and lakes. Shang et al. investigated the release from road sediments and found that the particle size distribution of the sediment had a significant effect on the nitrogen and phosphorus release characteristics [16]. Bubba et al. examined the relationship between phosphorus adsorbed in river sediments and sediment particle size and correlated particle size with phosphorus adsorption [17]. Antelo and Garcia reported that, at low pH, particulate matter exhibited a strong adsorption capacity for phosphorus with a gradual decrease in uptake at a higher pH, whereas the adsorption capacity of particulate matter for ammonia nitrogen ($NH_4^+$-N) and organic nitrogen increased with increasing pH [18,19]. Pan et al. have demonstrated that nitrogen and phosphorus release in sediments from a surface water river supply was higher than that from a reclaimed water supply [15]. Tang et al. showed that sediments in reclaimed water recharged streams had a higher risk of phosphorus release compared to surface water streams [20]. However, given the relatively short period of time that reclaimed water has been used, there are few comparative studies that address possible changes in nitrogen and phosphorus content and water quality due to the pipeline sediments after they enter surface water and reclaimed water.

In this study, we have undertaken a comparative analysis of nitrogen and phosphorus release by particles of different sizes in three kinds of water (deionized water, reclaimed water (RW) and surface (river) water (SW). The mass fraction and water characteristics have been evaluated. The TP and $NH_4^+$-N were measured with respect to reaction time, and a first-order kinetics model has been applied to quantify the release mechanism.

## 2. Materials and Methods

### 2.1. Raw Water and Pipe Sediments

The river water samples were collected from the upper reaches of Bahe River (39°57′49″ N, 116°30′29″ E), as illustrated in Figure 1. Bahe River is an important branch of the North Canal system, located in the Dongcheng and Chaoyang districts. It rises in the northeast and eventually merges with the Warm Elm River at Calendula Street. The main tributaries

include Beihe River, Liangma River and Beituchenggou. The main river length is 21.63 km, and the drainage area is 158.4 square kilometers. The main sources of replenishment are reclaimed water and natural surface runoff. The river is the main water transport and landscape corridor in Chaoyang District and also serves as an important flood channel in the central city.

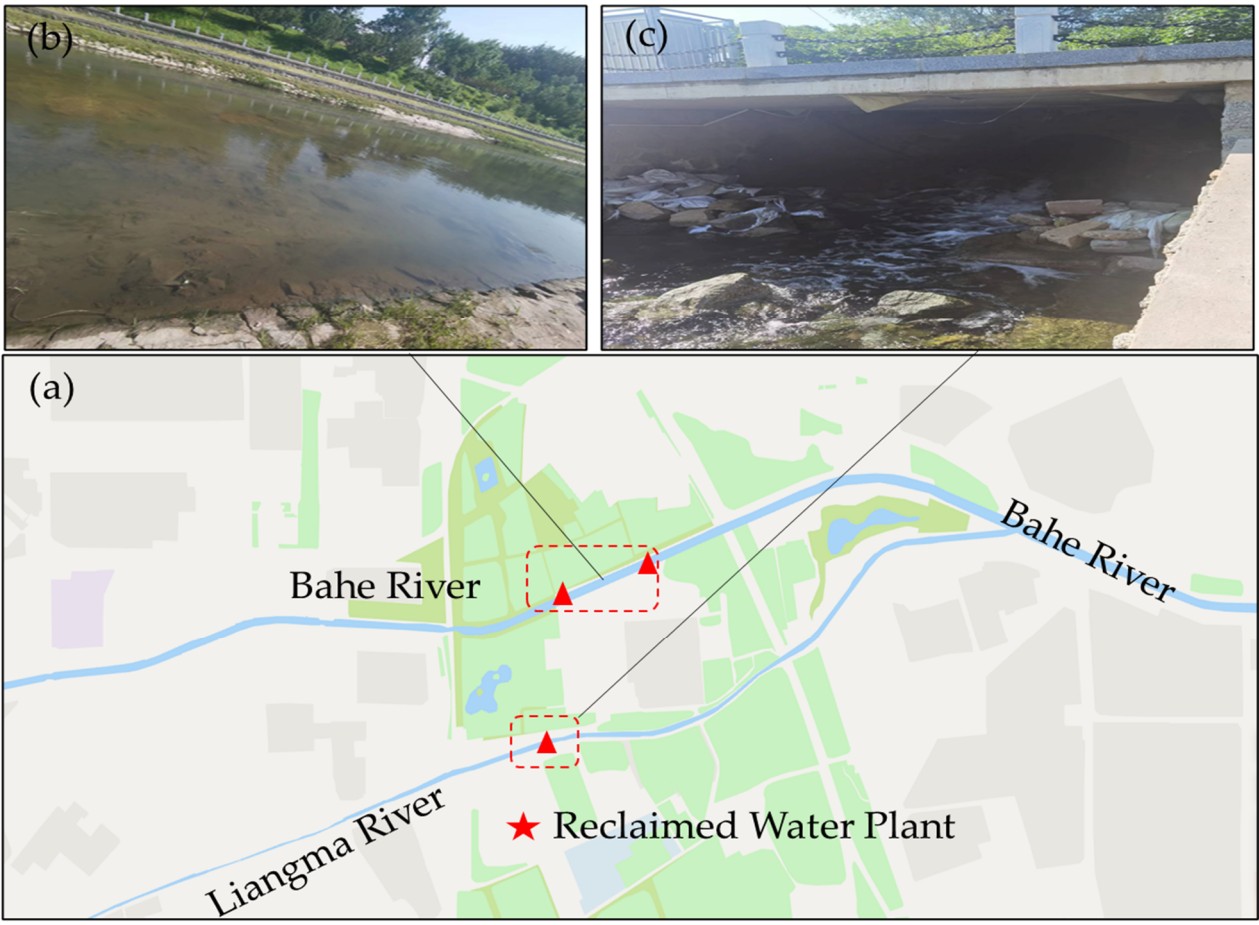

**Figure 1.** (**a**) The location of the sampling sites (the map resources from www.amap.com (accessed on 24 August 2022)); (**b**) photograph of the river water sampling sites; (**c**) photograph of the reclaimed water sampling sites.

The reclaimed water was collected from Beijing Jiuxianqiao Reclaimed Water Plant (treatment capacity: 200,000 cubic meters/day), located in Jiangtaiwa Village, Dongfeng Township, Chaoyang District, Beijing (39°58′5″ N, 116°27′48″ E), as shown in Figure 1c. The sewage is treated by an oxidation ditch activated sludge, and the reclaimed water is treated by a two-stage biological filter + sand filter/cloth filter. Part of the reclaimed water is used as supplementary water in the Bahe River Basin, and part is delivered to the reclaimed water users through a pipe network. Both river water and reclaimed water collections were completed on 10 August 2022. The river water samples were mixed samples collected at two river sampling sites; the reclaimed water samples were directly collected at the outlet. The water samples were all transferred into 2 L polyethylene sampling bottles and refrigerated and stored away from light before being sent to the laboratory for testing.

The sediments of the pipeline were collected from a rainwater pipeline in a campus in Beijing (39°44′58″ N, 116°16′59″ E) in August 2022. The whole pipeline forms a separate stormwater system without a domestic sewage mixed connection. A grapple type sediment sampler was used to extract pipeline sediment within different depths of the selected stormwater pipelines under aerobic conditions, and the samples were mixed samples that were collected at three times at the point. The mixed samples were sent to the laboratory

after the collection was completed and were freeze-dried with a freeze-dryer (FD-1A-50) at $-50$ °C and 0–10 Pa. The dried sediments were passed through a 16-mesh screen to remove impurities, and the final mass was 823.34 g. The dried sediments were screened using 42 mesh, 100 mesh and 200 mesh standard screens successively. The sediment samples were divided into four different particle sizes: >355 μm, 154–355 μm, 75–154 μm and <75 μm. The pipe sediments of different particle sizes were weighed separately on an electronic scale, and then the mass ratio of different particle sizes of pipe sediment was determined according to Formula (1) below. After screening, the sediments were stored at $-20$ °C before use.

$$C(\%) = \frac{m_i}{m} \qquad (1)$$

where $m_i$ is the mass of the pipe sediment with particle size i (g), and m is the total mass of the pipe sediment (g).

### 2.2. Experimental Design

Urban rivers are mainly recharged by river water and reclaimed water, and the contaminant content of pipeline sediments varies greatly among different particle sizes. Therefore, this study sets out to investigate the release characteristics of TP and $NH_4^+$-N from pipe sediment with different particle sizes in various types of water. The desorption experiment was conducted in a 3 L beaker with 40 g > 355 μm sediment dispersed under magnetic stirring in three water types (2 L): deionized water as the control group (CG), RW and SW. Samples (50 mL) were collected at intervals of 1, 3, 5, 7, 10, 20, 40, 60, 90, 120, 180, 240 and 300 min, and 13 samples were finally obtained. The samples were centrifuged and filtered through a 0.45 μm filtration membrane and then stored in 50 mL polyethylene bottles at 4 °C for a maximum of 24 h. The remaining sediments, in the size ranges of 154–355 μm, 75–154 μm and <75 μm, were sampled following the same protocol. Each experimental group was repeated once.

### 2.3. Analytical Techniques

The pH was measured using a portable multi-parameter water quality analyzer (American Hach HQ-40d, Hach Company (Loveland, CO, USA)). TP and $NH_4^+$-N were all analyzed using a spectrophotometer (LH-3BA(V12), Lianhua, China). The concentrations of TP and $NH_4^+$-N in water were determined by applying the ammonium molybdate spectrophotometric method (700 nm, 0.01–1.0 mg/L) and the salicylic acid spectrophotometric method (697 nm, 0.004–8.0 mg/L), respectively [15].

Due to the complexity of nitrogen and phosphorus components in particulate matter, some of the nitrogen and phosphorus will not be released after entering water, including, notably, calcium-bound phosphorus and some organic nitrogen, which is difficult to degrade. Therefore, we determined the alkali-hydrolyzable nitrogen, available phosphorus and ammonia nitrogen in sediments [21]. Alkali-hydrolyzable nitrogen includes inorganic nitrogen (ammonium nitrogen and nitrate nitrogen) and organic nitrogen that is readily hydrolyzed. Available phosphorus includes all water-soluble phosphorus, some adsorbed phosphorus, some slightly soluble inorganic phosphorus and mineralized organic phosphorus. The alkali-hydrolyzable nitrogen content and ammonia nitrogen content in sediments were measured by potassium chloride solution extraction and spectrophotometry. The quantity of available phosphorus in sediments was determined by sodium bicarbonate extraction and the molybdenum-antimony anti-spectrophotometric method.

### 2.4. Data Analysis

All data processing and plotting in the article were conducted in ORIGIN, and statistical analysis of data (ANOVA) was conducted in SPSS.

The release of TP and $NH_4^+$-N in sediments was calculated according to Formula (2):

$$Q = \frac{(C_e - C_0)V}{m} \qquad (2)$$

where $C_0$ is the initial concentration of river water and reclaimed water (mg/L), $C_e$ is the concentration after a given time interval (mg/L), V is the solution volume (L), and m is the weight of the pipe sediment (mg).

Previous studies have shown that the first-order kinetic equation more accurately describes the release of nitrogen and phosphorus from sediments [22]. Therefore, the first-order kinetic model is used to fit the nitrogen and phosphorus release process in this article [23]:

$$Q = E * \left(1 - e^{-bt-a}\right) \tag{3}$$

where E is the desorption amount at equilibrium (mg/g), b is the desorption rate constant, which reflects the degree of pollution, t is the desorption time (min), and a is the constant.

## 3. Results and Discussion

### 3.1. Pollution Characteristics of Water Samples and Pipe Sediments

The mass fraction of the four kinds of pipe sediment with different particle sizes is shown in Table 1. It should be noted that the highest proportion of particles was in the size range of 154~355 μm, accounting for a mass fraction of 36.04%. The sediment content of <355 μm was the lowest, with a mass fraction of 14.86%, and there was a near equivalence of particles <75 μm (22.44%) and in the 75–154 μm range (26.66%). Although most of the sediment in stormwater pipes comes from ground surface sediments, the sample particle size distribution differs from ground surface sediment size characteristics. Surface sediments typically show the largest proportion of particles less than 75 μm [24] as opposed to the predominant larger grain size (154~355 μm) found in pipe sediments (Table 1). This may be due to differences in the internal environment of the pipeline. Smaller particles are more likely to be washed away, while large particles are retained [25].

**Table 1.** Mass fraction of pipe sediments with different particle sizes.

| Particle Size Range (μm) | Proportion (%) |
|:---:|:---:|
| >355 | 14.86 |
| 154–355 | 36.04 |
| 75–154 | 26.66 |
| <75 | 22.44 |

The content of nitrogen and phosphorus pollution loads in the sediments of different particle sizes is shown in Figure 2a. It should be noted that the smaller the sediment particle size, the higher the content of nitrogen and phosphorus nutrients per unit mass. Sediments with particle sizes smaller than 75 μm exhibited the highest pollution load of alkali-hydrolyzable nitrogen, available phosphorus and ammonia nitrogen with percentages of 0.112 mg/g, 0.011 mg/g and 0.046 mg/g, respectively. These results are in agreement with a previous study [26]. At a smaller particle size, the specific surface area is larger, which should provide a greater adsorption capacity for nitrogen and phosphorus [27].

According to the mass fraction of sediments with different particle sizes in Table 1, The pollution load ratio was used to measure the contribution of sediments with different particle sizes to the total pollution of the pipeline. The measurement was calculated according to Formula (4):

$$GSF_{Load}(\%) = \frac{C_i \times GS_i}{\sum_{i=1}^{n} C_i \times GS_i} \tag{4}$$

where $C_i$ is the mass fraction of the pollutants in the particle size fraction i (mg/g), $GS_i$ is the mass fraction of the particle size fraction i (%), n is the number of particle groups, and n is taken as 4 in this paper.

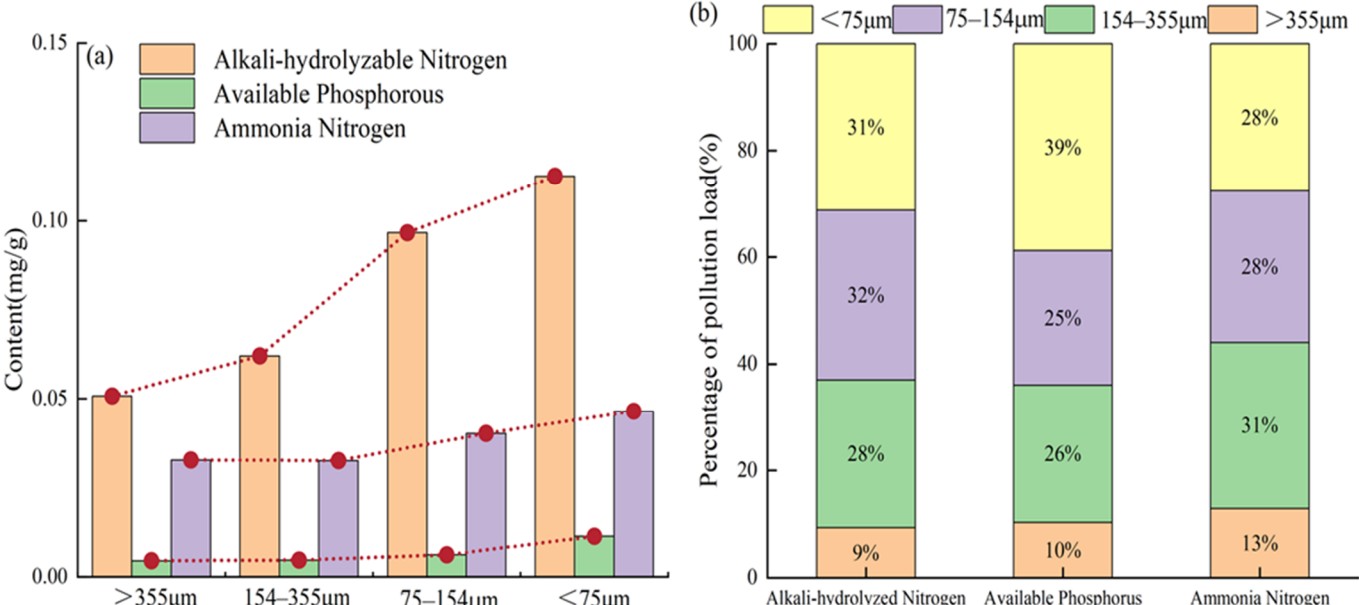

**Figure 2.** (**a**) Nitrogen and phosphorus mass fraction and (**b**) pollution load percentage in different particle size pipeline sediments.

The weighted nitrogen and phosphorus pollution loads are shown in Figure 2b. The maximum alkali-hydrolyzed nitrogen pollution load occurred in 75–154 μm pipeline sediments (32%), and the maximum available phosphorus and ammonia nitrogen pollution load was found in the sediments <75 μm (39%) and ranging 154–355 μm (31%). Although the pipe sediment with a smaller particle size had a higher nitrogen and phosphorus pollution load, as shown in Figure 2a, the difference in the mass fraction of the pipe sediment with different particle sizes resulted in a nitrogen and phosphorus pollution load with a different final weight. This is in agreement with a previous study [28].

### 3.2. Change in pH in Water before and after Reaction

Figure 3 shows the pH variation in the three types of water bodies before and after the addition of the sediment (denoted as BE and AF, respectively). It should be noted that the highest pH was recorded for SW before the addition of the sediment and that the pH of the RW was higher than the CG. This may be attributed to a depletion of $CO_2$ due to photosynthesis by phytoplankton in the RW [14]. After the addition of sediment, the pH of the three water bodies all increased. The starting pH of the CG was $7.1 \pm 0.2$, which was raised to $9.3 \pm 0.1$ following the inclusion of the sediment. In the case of the SW, the pH was increased from $8.9 \pm 0.1$ to $9.0 \pm 0.1$, while the increase in the pH of the RW was from $7.6 \pm 0.2$ to $8.1 \pm 0.1$ due to the sediment addition. The final pH followed the sequence CG > SW > RW, which is consistent with the research results reported by Hou et al. [10]. Compared with the CG, the pH variation in the case of the SW and the RW was significantly smaller. According to previous studies, this response can be ascribed to the presence of various ions in river water and reclaimed water that are formed by different acid-base buffer systems, such as carbonate, sulfate and Al(III) buffers, which played a role in buffering possible pH changes in water [29].

### 3.3. The Release Characteristics of TP

The relationship between TP release in different water bodies and pipe sediment particle sizes is shown in Figure 4. Initially, due to the large gap in TP concentration between the sediments and the accommodating waters, the TP release showed a marked increase, and, at extended reaction times, the release rate decreased to reach a plateau value representing equilibrium after 120 min. As illustrated in Figure 4a–d, it should be noted

that at equilibrium, the amount of TP released was higher in every case for the CG when compared with SW and RW. The amount of TP released in the CG was 1.06–1.56 times that of the SW and 1.23–2.44 times that of the RW. The results indicate that TP release was inhibited in both the SW and RW but especially in RW, which exhibited the lowest release (0.002 mg/g).

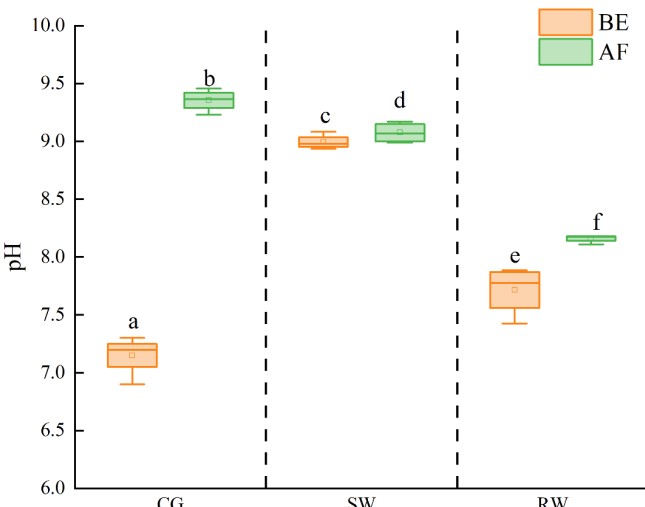

**Figure 3.** Changes in pH of different water bodies before and after pipeline sediment addition (different lower case letters above the columns indicate statistical differences at $p < 0.01$).

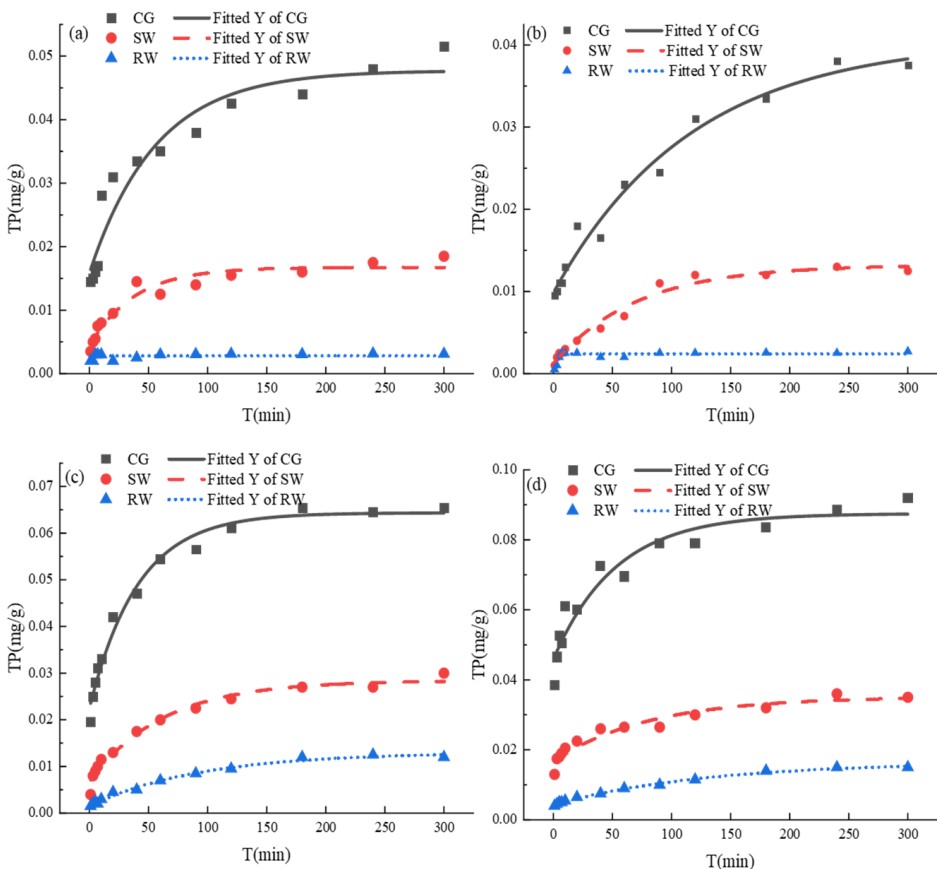

**Figure 4.** TP release kinetic curves for pipeline sediment with different particle sizes: (**a**) >355 μm; (**b**) 154–355 μm; (**c**) 75–154 μm; (**d**) <75 μm.

In order to further investigate the release kinetics of TP in sediments with different particle sizes, the first-order reaction kinetics model was used for data fitting, as shown in Table 2. It should be noted that the final TP release at equilibrium for the different sediment particle sizes decreased in the following order: E (<75 μm) > E (75–154 μm) > E (>355 μm) > E (154–355 μm). On the basis of $R^2$, the TP desorption from different particle sizes in different waters was in accordance with the primary reaction kinetic characteristics. Therefore, it can be inferred that the release of TP was mainly controlled by physical desorption [30].

**Table 2.** First-order reaction kinetic model fitting parameters for the desorption of TP from particulate matter.

|  | Classification | E (mg/g) | b | a | $R^2$ |
|---|---|---|---|---|---|
| | CG | 0.047 | 0.027 | 0.38 | 0.92 |
| >355 μm | SW | 0.016 | 0.026 | 0.25 | 0.94 |
| | RW | 0.003 | 0.022 | 0.19 | 0.93 |
| | CG | 0.041 | 0.018 | 0.28 | 0.98 |
| 154–355 μm | SW | 0.013 | 0.014 | 0.07 | 0.98 |
| | RW | 0.002 | 0.004 | 0.37 | 0.85 |
| | CG | 0.064 | 0.024 | 0.42 | 0.98 |
| 75–154 μm | SW | 0.028 | 0.015 | 0.25 | 0.97 |
| | RW | 0.013 | 0.010 | 0.12 | 0.98 |
| | CG | 0.087 | 0.018 | 0.73 | 0.93 |
| <75 μm | SW | 0.035 | 0.012 | 0.64 | 0.92 |
| | RW | 0.016 | 0.007 | 0.26 | 0.99 |

The suppression of TP release in SW and RW may be due to the higher TP concentration in SW and RW, which inhibited the ionic diffusion of phosphate in the particulate matter. Moreover, the variation of pH in the different water samples (as displayed in Figure 3) has established that the pH in SW and RW after the reaction was lower than that in the CG. At a lower water pH, the surface of the particulate matter exhibits a greater positive charge. The greater electrostatic attraction generated by the positive charge favors phosphate adsorption [31]. In addition, the lower pH impeded the binding of $Fe^{3+}$ and $Al^{3+}$ to $OH^-$, hindering the release of Fe-P and Al-P from the particulate matter. A previous study has also suggested that organic matter in reclaimed water can bridge with $Fe^{3+}$ and $Al^{3+}$ and other metal ions, combining with phosphorus to form stable organic-metal chelates, which favors the adsorption of phosphorus on particulate matter [32].

*3.4. The Release Characteristics of $NH_4^+$-N*

The release kinetics of $NH_4^+$-N from pipeline sediments with different particle sizes in different water bodies are presented in Figure 5. The release process of $NH_4^+$-N can be divided into two stages: a fast reaction and a slow reaction. The amount of $NH_4^+$-N released gradually approached an invariable value with increasing reaction time. The $NH_4^+$-N released (0.0188 mg/g) in RW from the pipeline sediments was higher than that in SW and the CG, as shown in Figure 5d. The release of $NH_4^+$-N in SW was also higher than that in the CG. This indicates that the release of $NH_4^+$-N in particulate matter was promoted in SW and RW, and the greater release of $NH_4^+$-N in RW can be linked to the lowest pH in RW. When the pH was raised to 7.0, it exceeded the zero charge of the submerged mud colloid, rendering the surface of the particulate matter negatively charged. The higher the pH, the more favorable the adsorption of particulate matter $NH_4^+$, thus inhibiting the release of $NH_4^+$-N into the surrounding water. At the same time, $H^+$ at the lower pH competes strongly for $NH_4^+$ in the mud colloid and contributes to desorption. It has also been shown that there is a large amount of organic matter in the reclaimed water and river

water, and the organic matter in the water can have adsorption competition behavior with $NH_4^+$, thus weakening the adsorption capacity of particulate matter on $NH_4^+$-N [33].

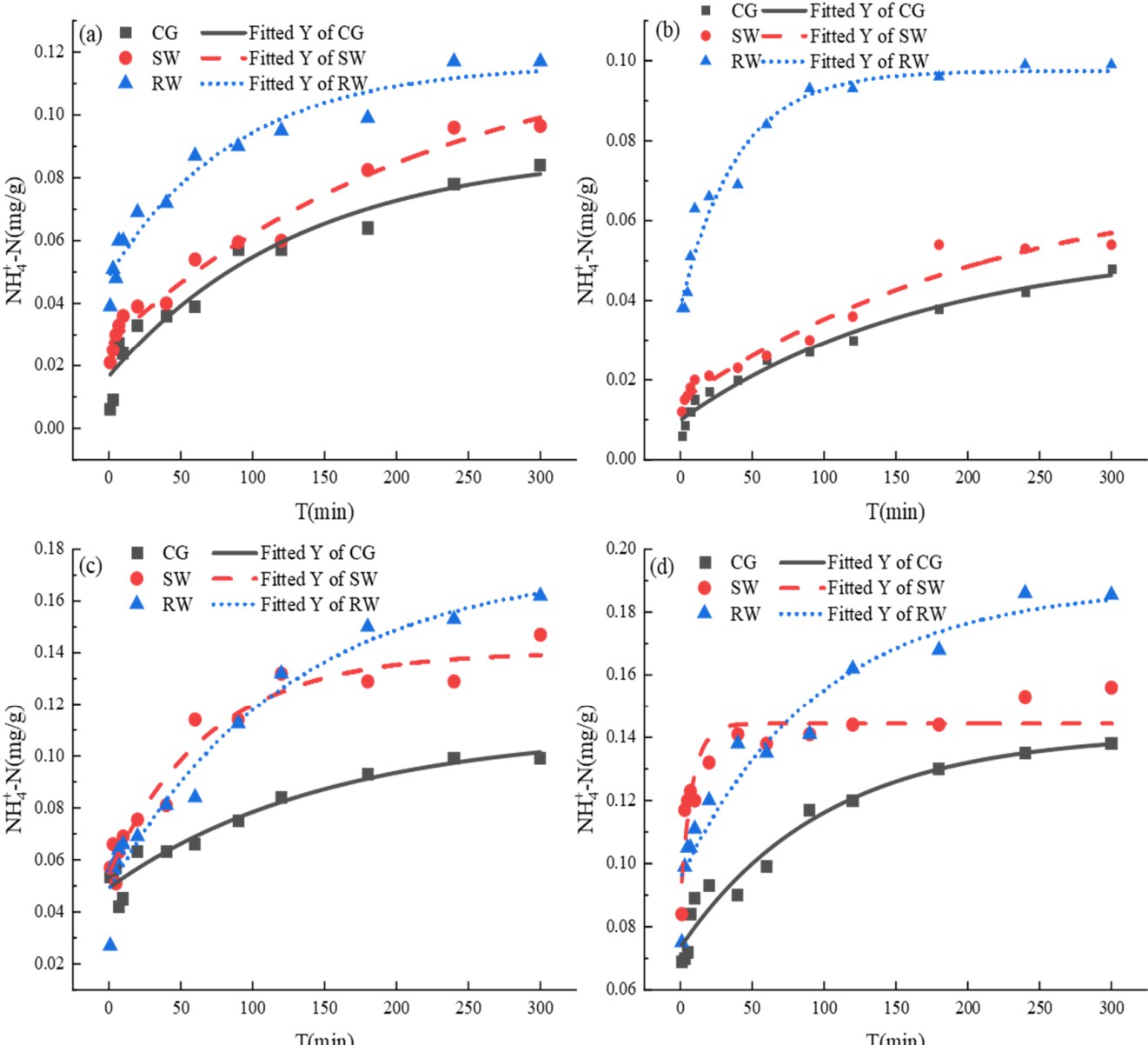

**Figure 5.** $NH_4^+$-N release kinetic curves for pipeline sediments with different particle sizes: (**a**) >355 μm; (**b**) 154–355 μm; (**c**) 75–154 μm; (**d**) <75 μm.

The first-order reaction kinetics model (Equation (2)) was used to fit the $NH_4^+$-N data; the fitting parameters are given in Table 3. It should be noted that the $NH_4^+$-N release at reaction equilibrium for different particle sizes of the sediments decreased in the following order: E (<75 μm) > E (75–154 μm) > E (>355 μm) > E (154–355 μm). According to $R^2$, the release of $NH_4^+$-N in the pipeline sediments conformed to the first-order reaction kinetics model with a high correlation coefficient. Therefore, the release of $NH_4^+$-N was also mainly controlled by physical desorption [23].

**Table 3.** The first-order reaction kinetic model fitting parameters for the desorption of $NH_4^+$-N from particulate matter.

| | Classification | E (mg/g) | b | a | $R^2$ |
|---|---|---|---|---|---|
| | CG | 0.0088 | 0.008 | 0.21 | 0.94 |
| >355 μm | SW | 0.0115 | 0.004 | 0.16 | 0.97 |
| | RW | 0.0116 | 0.011 | 0.54 | 0.95 |
| | CG | 0.0054 | 0.005 | 0.20 | 0.98 |
| 154–355 μm | SW | 0.0071 | 0.004 | 0.23 | 0.96 |
| | RW | 0.0097 | 0.025 | 0.51 | 0.96 |
| | CG | 0.0109 | 0.006 | 0.59 | 0.94 |
| 75–154 μm | SW | 0.0140 | 0.014 | 0.49 | 0.95 |
| | RW | 0.0174 | 0.007 | 0.32 | 0.96 |
| | CG | 0.0141 | 0.009 | 0.73 | 0.96 |
| <75 μm | SW | 0.0144 | 0.125 | 0.93 | 0.85 |
| | RW | 0.0188 | 0.012 | 0.70 | 0.94 |

*3.5. Percentage of TP and $NH_4^+$-N Released by Pipe Sediments of Different Particle Sizes*

Taking the maximum released amounts of TP and $NH_4^+$-N recorded in Tables 2 and 3 and the content of available phosphorus and ammonia nitrogen in the sediments of different particle sizes, the percentage release was calculated and is presented in Figure 6. It should be noted that the sediment in the size range 75–154 μm exhibited the highest percentage of TP and $NH_4^+$-N release into the three water samples. This may be due to the relatively high concentration of nitrogen and phosphorus released from the 75–154 μm pipeline sediments, which, on the other hand, have a lower concentration of nitrogen and phosphorus pollutants fused to them compared to the <75 μm pipeline sediments. The percentage release of both TP and $NH_4^+$-N was lowest in the 154–355 μm sediment size range. In addition, the percentage of TP release from the 75–154 μm sediment was the highest in the CG (34.53%), while the percentage of $NH_4^+$-N release from the 75–154 μm particles was the highest in RW (43.51%). Therefore, the 75–154 μm pipe sediments exhibited the greatest risk of contamination. The percentage release of TP in the CG was 2.01–3.08 times that in the SW and 5.21–15.97 times that in the RW.

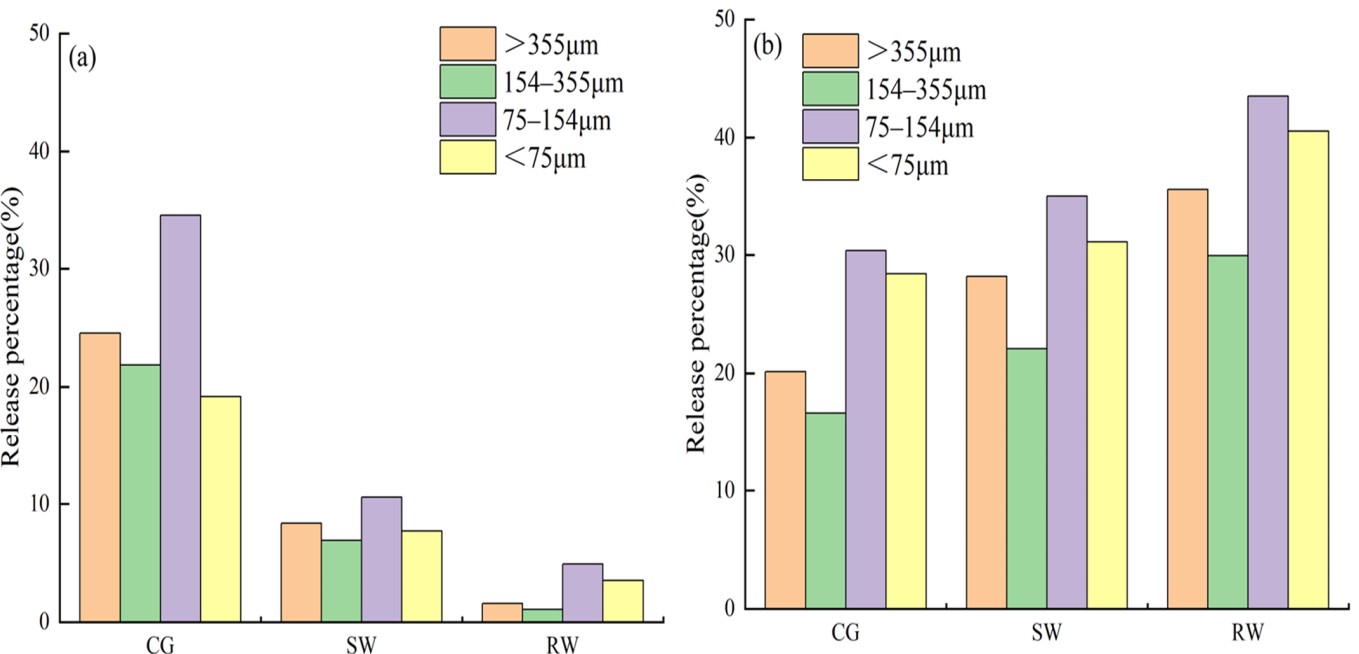

**Figure 6.** The percentage release of (**a**) TP and (**b**) $NH_4^+$-N in sediments with different particle sizes.

## 4. Conclusions

This study has explored nitrogen and phosphorus release characteristics from sediments with different particle sizes after entering different water bodies. The results support the following conclusions. The mass fraction of the 154–355 μm particles in the pipeline sediments was the largest. Although the sediment with a smaller particle size can accumulate more pollutants, the pollution load of the weighted $NH_4^+$-N and TP was the highest in the sediment of the 75–154 μm size range. Both the release of TP and $NH_4^+$-N from pipeline sediments conformed to the first-order reaction kinetics model. The release of TP was inhibited in the SW and the RW to varying degrees, and the RW exhibited a stronger inhibition of TP release. $NH_4^+$-N release was promoted in the SW and the RW, and the RW showed a stronger promotional effect with respect to $NH_4^+$-N. The percentages of TP and $NH_4^+$-N release were highest for the sediments in the size range of 75–154 μm. Therefore, attention should be paid to the characteristics of runoff pollution into different water bodies as possible recharging sources in order to control and manage urban water pollution. This study is a significant guide and reference for the regulation of water quality in urban rivers with different recharge sources, during and after rainfall in the future, to maintain a high standard of urban water environment quality operations. The seasonal variation of surface water environmental quality in consideration of nitrogen and phosphorus release and the long-term pollution characteristics of runoff pollution after they enter different water bodies, as well as the associated ecological risks, are the next steps for this research.

**Author Contributions:** Data curation, J.S.; writing—original draft preparation, J.S. and C.X.; writing—review and editing, J.S. and C.X.; supervision, J.L. and W.W. All authors have read and agreed to the published version of the manuscript.

**Funding:** This research was funded by the National Natural Science Foundation of China, grant number 2021YFC3200701.

**Institutional Review Board Statement:** Not applicable.

**Informed Consent Statement:** Not applicable.

**Data Availability Statement:** Not applicable.

**Conflicts of Interest:** The authors declare no conflict of interest.

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
