# Peer review of "Nitrogen and Phosphorus Release Characteristics of Pipeline Sediments on Entering Different Water Bodies"

_water, doi:10.3390/w15101903_

Round 1
Reviewer 1 Report
This paper details Nitrogen and phosphorus release characteristics of pipeline sediments on entering different water bodies.
It is an interesting and complete study, with high applicability.
This paper can be published in Water.
Author Response
Dear Reviewers, Thank you very much for your comments on our research and for your recognition.
Reviewer 2 Report
No comments for authors
Author Response

(The authors gave the same response as above.)

Reviewer 3 Report
The article "Nitrogen and phosphorus release characteristics of pipeline sediments on entering different water bodies" provides valuable scientific information about water reclamation issues. However, the authors need to address the following comments-
Water eutrophication should be replaced by 'eutrophication'.
Appropriate and uniform referencing conventions must be followed throughout the entire article.
Figure 1: The source of the map needs to be mentioned in the article.
The coordinates of the sampling points need to be mentioned in the article.
The authors should provide justification for not measuring the total nitrogen along with ammonium nitrogen.
Did the authors take existing surface water qualities along with the seasonal variations into account? It is possible that the chemical characteristics of the surface water can affect the release of nutrients coming from the runoff.
The article needs grammatical formatting.
Author Response
Dear reviewrs,Thank you and reviewers very much for the comments on this manuscript。According to the comments from the reviewers, we carefully checked and completed the necessary corrections in the submitted revised manuscript. We believe that our manuscript has improved by addressing to your comments and hope the manuscript can be accepted for publication.

Reviewer 4 Report
Comments on the author:
The manuscript is the release characteristics of nitrogen and phosphorus in pipeline sediments. This is an important topic related to urban water environment management, which has certain research value and practical significance, but also some problems and shortcomings that need to be revised and improved.
1、In the introduction part, it should increase the review of relevant research at home and abroad, highlights the innovation and contribution of this research, and clearly puts forward the research purpose and hypothesis. Increase the background introduction and significance of reclaimed water and surface water as urban water environment recharge water sources to increase the urgency and pertinence of your research questions.
2、In the material and method section, the basis and principle of the experimental design should be explained, such as why sediments of different particle sizes and different types of water bodies should be selected, and why the first-order kinetic model is used to fit the nitrogen and phosphorus release process.
3、In the part of materials and methods, the control methods of experimental error and repeatability should be given, such as the number of samples and repetition times of each experimental group, as well as the software and methods of data processing and analysis.
4、In the experimental method section, explain in detail how you select and collect pipeline sediment samples, and how you determine the mass ratio of sediments of different particle sizes to increase your experimental design and data reliability.
5、In the results and discussion section, the experimental results should be fully explained and analyzed, and the influencing factors and mechanisms should be discussed, such as why the release of nitrogen and phosphorus in reclaimed water is inhibited or promoted, and why 75-154μm sediments have the highest percentage of nitrogen and phosphorus release?
6、In the discussion section, the in-depth analysis and discussion of the influencing factors of the experimental results should be increased, such as the influence of other ions ( such as calcium, magnesium, etc. ) in the water body on the adsorption or desorption of nitrogen and phosphorus, and the mechanism of organic matter in reclaimed water and surface water on the migration and transformation of nitrogen and phosphorus.
7、In the conclusion part, the main findings and significance of this study should be summarized, and the guiding role and application value of this study on urban water environment management should be pointed out.
Moderate editing of English language
Author Response

(The authors gave the same response as above.)

Reviewer 5 Report
Manuscript ID: Water - 2373195
General comments:
1. This paper aims to highlight one of the most important water quality concern, having ensure water
systems protection; the release of N and P from sediments to water. Comparing the effects on river fresh
water and reclaimed water (an important source of water to face water scarcity).
2. The authors have done a lot of work and obtained good quality data.
Specific comments:
3. Line 82 Section “Materials and methods”: Needs a major revision. Some more information needs to be
presented: 2.1” Raw water and pipe sediments “The water samples (SW and RW) were collected once
only, on which date? mixed sample or simple sample? How exactly were the sediment samples taken and
processed? What was the sample size? Are they all from the same depth? Were they collected anoxically?
Are the sediments oxic or anoxic? Which was the drying procedure (conditions)?
2.3. “Analytical Techniques”. Complete the information concerning the instrumentation (apparatus),
include the wavelengths and detection limits for TP and NH4-N. Whole methods need to be referenced
(citation).
2.4. “data analysis” Statistical analyses need to be included.
4. Line 139-140: m is the weight of the pipe sediment in g not in L.
5. Line 181 “Change of pH in water before and after reaction”: A statistical analysis need to be done (as
ANOVA and Tukey test), and the statistical results present in the tables and figures throughout this
manuscript. Lacking statistical results would make readers doubt whether those differences are
significant or not (to apply also TP and NH4-N), namely for the pH changes between SW and RW (it doesn´t
seem significantly). For the same reason, it is hard to justify the conclusion statements in the paper. I’d
suggest the authors add statistical analysis results to all tables and figures and re-check/re-write the text
according to the results.
Author Response

(The authors gave the same response as above.)

Round 2
Reviewer 4 Report
The manuscript has improved by addressing to my comments and the manuscript can be accepted for publication.
Minor editing of English language required